# New Breakfast Cereal Developed with Sprouted Whole Ryegrass Flour: Evaluation of Technological and Nutritional Parameters

**DOI:** 10.3390/foods12213902

**Published:** 2023-10-25

**Authors:** Cristiane Teles Lima, Tatiane Monteiro dos Santos, Nathália de Andrade Neves, Alicia Lavado-Cruz, Luz Maria Paucar-Menacho, Maria Teresa Pedrosa Silva Clerici, Sílvia Letícia Rivero Meza, Marcio Schmiele

**Affiliations:** 1Institute of Science and Technology, Federal University of Jequitinhonha and Mucuri Valleys (UFVJM), Diamantina 39100-000, Brazil; cristiane.lima@ufvjm.edu.br (C.T.L.); tatiane.santos@ufvjm.edu.br (T.M.d.S.); nathalia.neves@ict.ufvjm.edu.br (N.d.A.N.); 2Departamento de Agroindustria y Agrónoma, Universidad del Santa (UNS), Nuevo Chimbote, Ancash 02712, Peru; 201612019@uns.edu.pe (A.L.-C.); luzpaucar@uns.edu.pe (L.M.P.-M.); 3Departament of Food Science and Nutrition, University of Campinas (UNICAMP), Campinas 13083-862, Brazil; mclerici@unicamp.br; 4Department of Agroindustrial Science and Technology, Federal University of Pelotas (UFPel), Pelotas 96160-000, Brazil; silvialrmeza@gmail.com

**Keywords:** bioprocess, cereal, extrusion, germination, healthiness, phytochemicals

## Abstract

Ryegrass is one such cereal that has been underutilized in human nutrition despite its high nutritional and functional value due to the presence of phytochemicals and dietary fibers. Exploiting ryegrass for human consumption is an exciting option, especially for countries that do not produce wheat, as it is easily adaptable and overgrows, making it economically viable. This study evaluated the nutritional content of γ-aminobutyric acid and bioactive compounds (total soluble phenolic compounds) and the physicochemical and technological properties of partially substituting maize flour (MF) with sprouted whole ryegrass flour (SR) in developing extrusion-cooked breakfast cereals. A completely randomized design with substitutions ranging from 0 to 20% of MF with SR was employed as the experimental strategy (*p* < 0.05). Partial incorporation of SR increased the content of γ-aminobutyric acid and total soluble phenolic compounds. Using sprouted grains can adversely affect the technological quality of extruded foods, mainly due to the activation of the amylolytic enzymes. Still, ryegrass, with its high dietary fiber and low lipid content, mitigates these negative effects. Consequently, breakfast cereals containing 4 and 8% SR exhibited better physicochemical properties when compared to SR12, SR16, SR20, and USR10, presenting reduced hardness and increased crispness, and were similar to SR0. These results are promising for ryegrass and suggest that combining the age-old sprouting process with extrusion can enhance the nutritional quality and bioactive compound content of cereal-based breakfast products while maintaining some technological parameters, especially crispiness, expansion index, water solubility index, and firmness, which are considered satisfactory.

## 1. Introduction

Breakfast cereals are among the ready-to-eat products that have been gaining market share and are widely consumed worldwide. This effect may be associated with claims such as convenience, rapid and easy preparation, a lack of time for consumers to prepare daily meals, and the excellent availability of products with recognized nutritional and physiological properties [1].

The most common raw materials for producing breakfast cereals are wheat and maize or corn, used mainly as refined flours, where nutrients such as dietary fibers, vitamins, and minerals are present at low concentrations. Demand for new products is rising to align with public health policies and promote adequate nutrition, all within the framework that combines health and sustainability. Ingredients rich in soluble and insoluble dietary fibers and phytochemicals are related to promoting consumer health, where insoluble fibers slow down the gastrointestinal tract and soluble fibers reduce the rate of glucose absorption [2]. There has been a growing interest in bioactive compounds in recent years due to their capacity to neutralize free radicals or reactive oxygen species within biomolecules. This plays a role in mitigating oxidative stress, ultimately contributing to health promotion and preventing noncommunicable diseases. (NCDs). Among the commonly known bioactive compounds, the phenolic compounds are divided into phenolic acids (hydroxybenzoic acids—gallic, protocatechuic, *p*-hydroxybenzoic, vanillic, and syringic acids—and hydroxycinnamic acids—*p*-coumaric, ferulic, caffeic, sinapic, chlorogenic, and cinnamic acids), flavonoids (flavonols, flavanols, flavones, flavanones, anthocyanidins, and isoflavonoids), and non-flavonoids (stilbenes, chalcones, coumarins, lignans, and tannins). These compounds differ in chemical structure (aromatic rings and organic carboxylic acid), the number of functional chemical groups (hydroxyls), and distinct polarities. Phenolic compounds are considered secondary metabolites that plants use as a defense, resulting from biotic factors (insects, microorganisms, rodents, and birds) and abiotic factors (humidity, temperature, ultraviolet solar radiation, and nutrients). These compounds have antioxidant, immunomodulatory, antimutagenic, and anti-inflammatory properties [3] and are also associated with palatability and food intake.

Ryegrass (*Lolium multiflorum* L.), belonging to the *Poaceae* family, is a high-productivity crop and has not yet been used in human food. This cereal is a potential candidate for innovation and to increase the worldwide supply of food due to its morphological, agronomic, and nutritional characteristics, highlighting the easy dispersion and adaptation to low and medium fertility soils and pest resistance, among others, and having easy adaptation in countries with partial or total territory located below the Capricorn or above the Cancer Tropics [4].

As a form of grain processing, sprouting is a bioprocess in which the induction for bioconversion and/or release of bioactive compounds occurs, which is a very advantageous biotechnological process that is easy and low-cost and has contributed to improvements in the nutritional, technological, and sensory properties of cereal products [5]. Since ryegrass is a dressed caryopsis with smaller morphometric characteristics than rice grains, germination can favor husking during grain processing.

In general, sprouting biotechnology is characterized by the visible development of the radicle. Sprouting is only possible due to the metabolic processes that occur on macronutrients for the growth of radicle [6]. Among the main macronutrients found in cereals are starch and structural proteins (prolamine and glutelin), which are found in endosperm. The pericarp has dietary fibers and minerals, and the germ is rich in lipids, reserve proteins (albumin and globulins), sugars, minerals, and vitamins [7]. The chemical composition of ryegrass concerning proteins draws attention because it has about 12.11–12.83% of proteins, most of them composed of glutelins (28.87–33.09%) and albumin (29.81–31.88%), followed by prolamines (16.18–19.47%) and globulins (13.11–14.79%) [8].

Thus, through the sprouting process, the hydrolysis of macronutrients occurs by the action of endogenous enzymes, resulting in low-molar-mass molecules [9,10]. There is also the release and bioconversion of bioactive compounds that are usually complexed with dietary food, and, by the action of the esterases and cellulases, these phytochemicals are released, favoring bioavailability and bioaccessibility. Cereal germination is seen as a possibility for the development of nutritious breakfast cereals as well as enhancing the content of bioactive compounds.

Cooking extrusion is the most used technological process for producing breakfast cereals and/or expanded extrudates, and it involves a set of unit operations such as mixing, cooking, kneading, shearing, and shaping [11]. During the extrusion cooking of cereal flours or grits, starch gelatinization and protein denaturation occur, forming a highly viscous and amorphous paste. This paste significantly influences the microstructure, chemical characteristics, shape, and texture of the food material [12]. Meza et al. [13] developed gluten-free breakfast cereals using nonconventional flours of red and black rice as the primary ingredients through thermoplastic extrusion processing technology. They obtained gluten-free breakfast cereals from 100% pigmented rice flour with satisfactory sensory characteristics. Based on the results, the authors concluded that extrusion is a promising technology for creating innovative extruded products. Furthermore, thermoplastic extrusion is an advantageous processing technology for product development due to its versatility, low cost, and environmental friendliness, generating minimal waste.

Based on the preceding, this is the first study with reports of the application of ryegrass in products for human consumption. This study aimed to investigate the changes induced by the sprouting process of ryegrass seed and the performance of sprouted ryegrass whole flour on the physicochemical, technological, and nutritional properties of ready-to-eat breakfast cereal. The proposal of combining two whole flours (maize and ryegrass) aimed to increase the protein and dietary fiber content and, in the case of SR, to enhance the presence of phenolic compounds resulting from the sprouting process while minimizing the detrimental effects on technological functionality provided by the substitution of MF.

## 2. Materials and Methods

### 2.1. Materials

The seeds of the RG-LE1963 ryegrass variety of Uruguayan origin were acquired from the market in the city of São Lourenço do Sul (Brazil). The yellow dent maize grain was purchased from the market in the city of Nuevo Chimbote (Peru). The project was registered under number ADE0BDA in the National System for the Management of Genetic Heritage and Associated Traditional Knowledge (SisGen) of the Ministry of the Environment of the Federative Republic of Brazil. All chemical reagents used to develop this work were of analytical grade and possessed the required purity per the chosen analytical methodology.

### 2.2. Methods

#### 2.2.1. Ryegrass Sprouting

Ryegrass seeds (400 g) were previously sanitized with 2.4 L of NaClO 250 ppm (*v*/*v*) for 30 min. Grains were washed with distilled water to reach a neutral pH and then soaked in distilled water (1:6 *w*/*v*) for 4 h at room temperature (~20 °C). The ryegrass seeds were placed in polyethylene trays (0.14 m^2^ surface area), with the bottom and top of the trays covered by a layer of cotton, and the hydrated seeds of ryegrass were placed inside, separated by an intermediate layer of paper (0.044 g·m^−2^). Each cotton layer was moistened with 100 mL of distilled water.

The trays were placed in a BOD TF-33A (Telga, Belo Horizonte, Brazil), and the sprouting process was conducted in the dark at a temperature of 20 °C for 95 h and sprayed with distilled water every 12 h. After this process, the sprouted seeds were placed in perforated stainless-steel trays (0.14 m^2^ surface area) and dried in a forced air circulation and renewal oven (1 m·s^−1^) TE-394/1 (Tecnal, Piracicaba, Brazil) at 45 °C for 20 h. The sprouted seeds were packaged in bioriented polypropylene packaging, stored under refrigeration (4 °C), and protected from light for subsequent grinding [6].

#### 2.2.2. Obtaining the Whole Flours of Unsprouted Ryegrass, Sprouted Ryegrass, and Maize

The unsprouted (USR) and sprouted (SR) ryegrass seeds were ground into flour using a knife mill MA1680 (Marconi, Piracicaba, Brazil) with a 0.3 mm sieve (Figure 1A,B). The yellow dent maize grains were processed into whole maize flour (MF) using a hammer mill MDNT-60XL (Torrh, Junín, Peru) with a 0.5 mm sieve. The difference in sieve openings occurred because the raw materials were ground in different mills and countries, allowing the use of the available sieves.

#### 2.2.3. Centesimal Composition of Whole Ryegrass and Maize Flour

The flours were evaluated for their centesimal composition by determining the levels of proteins, ether extract, and ash according to the methods 46-13.01 (N = 6.25), 30-25.01, and 08-01.01, respectively, as established by the American Association of Cereal Chemists International (AACCI, 2010) [14]. The contents of digestible carbohydrates (sugars and starch) were determined by method 982.14, and total dietary fiber was determined by method 978.10 of the Association of Official Analytical Chemists (AOAC, 2019) [15]. All analyses were performed in triplicate, and the results are expressed as g·100 g^−1^ (d.b.).

#### 2.2.4. Experimental Design, Test Preparation, and Thermoplastic Extrusion Processing to Produce Breakfast Cereals

The tests for the experiments were developed using a completely randomized design (CRD), and the extrusion processing was carried out at the Departamento de Agroindustria y Agrónoma from the Universidad del Santa in Nuevo Chimbote (Ancash, Peru). SR partially replaced the MF at proportions of 0% (SR0), 4% (SR4), 8% (SR8), 12% (SR12), 16% (SR16), and 20% (SR20), and by 10% of USR (USR10) (*w*/*w*).

The samples were preconditioned to 15% moisture content by slowly and gradually adding potable water (~20 °C) and mixing for 5 min at low speed in a K45SS planetary mixer (Kitchen Aid Professional, St. Joseph, MO, United States of America). They were then transferred to bioriented polypropylene packaging, vacuum-sealed, and stored overnight under refrigeration (7 °C) for moisture equilibrium.

The extrusion cooking (a 2nd-generation extrusion process that enables the final product to be ready-to-eat without the need for additional heat treatment, such as toasting, flaking, or frying) was carried out using a Labor PQ DRX-50 intermeshing co-rotating twin-screw extruder (Inbramaq, Ribeirão Preto, Brazil), with a barrel consisting of seven heating zones, with temperatures ranging from 30, 45, 55, 75, 95, 105, and 115 °C from the feeding zone to the die, respectively. The feeding rate was set at 107 g·min^−1^, and the screw speed was set at 330 rpm, with a screw length of 870 mm and a diameter of 32 mm (L/D = 27). A circular die with one orifice (6 mm in diameter) was used. The samples were cut using a rotary knife, cooled, and stored in bioriented polypropylene packaging at room temperature (~20 °C), protected from light.

#### 2.2.5. Physico-Chemical Properties of Breakfast Cereals

The breakfast cereals produced were evaluated for the following parameters: in its entirety form for radial expansion index (REI), bulk density (BD), instrumental color, instrumental of dry and bowl-life texture, and in milled form for water absorption index (WAI), water solubility index (WSI), γ-aminobutyric acid (GABA), and total soluble phenolic compounds (TSPC).

##### Expansion Index, Bulk Density

REI and BD were analyzed according to the methodology described by Paucar-Menacho et al. [11]. The REI was calculated by dividing the diameter of the breakfast cereal (mm) by the diameter of the extrusion cooking die (mm) used in the process. Measurements were taken with ten repetitions using a professional 150 mm analog caliper (Loyal, Poços de Caldas, Brazil).

For BD, measurements of the dimensions of the extrudates were obtained using a professional 150 mm analog caliper, and the weight of samples was determined on an analytical balance AUY220 (Shimadzu, Tokyo, Japan), with ten repetitions. The BD data were obtained according to Equation (1), and the results were expressed in g·cm^−3^.
BD (g·cm^−3^) = *w*/π*l*(*d*/2)^2^(1)
where *w* is the weight (g), *l* is the length (cm), and *d* is the diameter (cm) of the sample.

##### Instrumental Color

For instrumental color analysis, a CM-5 Konica spectrophotometer colorimeter (Minolta, Chiyoda, Japan) was used, with four repetitions for each raw material and breakfast cereals, and the results were obtained using the CIE *L***a***b** system. The total color difference (ΔE) was defined by the numerical comparison between the test made with MF only (SR0) and the other tests. Equation (2) was used to determine ΔE between the three coordinates. The test conditions were illuminant D65, observation angle of 10°, and calibration mode in RSIN (Reflectance Specular Included).
ΔE = [Δ*L**^2^ + Δ*a**^2^ + Δ*b**^2^]^1/2^(2)

##### Instrumental Texture of Dry Breakfast Cereals and after Immersion in Refrigerated Whole Milk (Bowl-Life)

The breakfast cereals were dehydrated in a TE-394/1 oven with air circulation and renewal (1 m·s^−1^) (Tecnal, Piracicaba, Brazil) at 80 °C for 2 h to achieve a moisture content below 6%, as established for breakfast cereals. The texture of dry breakfast cereals was evaluated as described by Paucar-Menacho et al. [11].

The bowl-life analysis was conducted as described by Oliveira et al. [16], with few modifications. In summary, the breakfast cereals were immersed in whole milk (3% lipids) at 9 °C for 3 min in a ratio of 1:3 (breakfast cereals:whole milk). Subsequently, the breakfast cereals were drained for 10 s using a household sieve and then subjected to texture analysis.

The extrudates were evaluated using a TAXT Plus texture analyzer (MicroStable Systems, Halsemere, Australia) adopted with an HDP/90 platform and a Warner Bratzler probe with a thickness of 1 mm and a 60° angular blade. The tests were conducted in compression mode under the following operating conditions: pre-test and test speeds of 0.5 mm·s^−1^, post-test speed of 10 mm·s^−1^, a distance of 25 mm, and a detection threshold of 0.049 N. Eight repetitions were performed for each dry and bowl-life test. The extrudates were evaluated for hardness, rupture strength, and crispness, where hardness was defined as the peak of the maximum force (N) from the first compression required to break the sample, rupture force (rupture resistance) was defined as the area under the curve (N·s), and crispness was determined by the total number of force peaks measured along the curve.

##### Water Absorption Index and Water Solubility Index

The samples were ground in a TE-350 ball mill (Tecnal, Piracicaba-SP, Brazil) for 4 min, and WAI and WSI were determined according to the methodology described by Schmiele et al. [17]. In Falcon tubes with a capacity of 15 mL, 1.0 g of the sample was suspended in 10 mL of distilled water at 25 °C for 30 min, with manual stirring every 5 min. Subsequently, the samples were subjected to phase separation in a Fanem Baby | Centrifuge (Tecnal, Piracicaba, Brazil) for 10 min at 3200× *g*. The supernatant was collected in a pre-weighed Petri dish. The water was evaporated in a TE-394/1 oven with air circulation and renewal at 85 °C overnight, and then final dehydration was performed at 2 h at 105 °C without air renewal. The analysis was performed in triplicate for the raw materials and breakfast cereals, and the results were expressed in g of gel per g of sample for WAI in the precipitated and as a percentage for WSI in the supernatant on a dry basis.

##### γ-Aminobutiric Acid

The GABA analysis was conducted on raw materials and breakfast cereals as described by Genčić et al. [18] with modifications. GABA was extracted from previously ground samples using a ball mill, and the extraction was performed with 75% ethanol (*v*/*v*), using 1.0 g of samples for 15 mL of extraction solution. Falcon tubes were placed on an SL 180/DT orbital shaker (Solab, Piracicaba, Brazil) for 1 h at 200 rpm and centrifugated at 9600× *g* at 4 °C for 15 min in a Sorvall ST8 Centrifuge (Thermo Fisher Scientific centrifuge, Waltham-MA, Estados Unidos), with the supernatants collected volumetric flask and the final volume adjusted to 20 mL with extraction solvent. The GABA present in the supernatant was derivatized to DNS-GABA using 100 µL of the extract, 200 µL of dansyl chloride (10 mM solution in acetonitrile (ACN)), and 200 µL of 5 M sodium carbonate buffer (pH 10). The solutions were subjected to low-frequency ultrasound in a CBU/100/3LDG ultrasonic bath (Planatec, São Paulo, Brazil) (40 kHz/100 W) for 3 min, followed by heating at 65 °C for 25 min in a SL-150 static water bath (Solab, Piracicaba, Brazil).

The identification and quantification of GABA were performed using a 1260 Infinity High-Performance Liquid Chromatography (HPLC) system (Agilent Technologies, Santa Clara, CA, United States of America) equipped with a quaternary pump and a G1314B diode array detector (DAD). The standard used for analysis was γ-aminobutyric acid (Merk, >99% purity). Chromatographic separations were carried in a Zorbax Eclipse C_18_ column (4.6 × 100 mm, 3.5 µm). The mobile phase consisted of sodium phosphate buffer (0.05 M, pH 7.2) and ACN with a flow rate of 1.0 mL·min^−1^. The column was initially eluted with a linear gradient in which the amount of ACN was gradually increased—14–20% ACN (0–7 min), 20–22% ACN (7–10 min), 22–25% ACN (10–15 min). Then, ACN was maintained at 85% for 3 min, followed by a linear gradient in which ACN decreased from 85 to 14% (25 to 26 min). All mobile-phase solvents were of HPLC grade. The injection volume was 10 µL, and detection was carried out at 254 nm. The quantification of GABA in the samples was performed using the GABA calibration curve method. External calibration was carried out using standard solutions of GABA in the linear range of six points between 0 and 184 mg·L^−1^) based on integrating the corresponding peak area (y = 29,245x + 30,702; r = 0.9994). The results were expressed as mg of GABA per 100 g of sample on a dry basis.

##### Total Soluble Phenolic Compounds

The TSPC content of the raw materials in the sanitization and maceration water and for breakfast cereals was performed according to Lima et al. [19]. In summary, for raw materials and ground breakfast cereals, an aliquot of 200 mg was weighed into 2000 μL Eppendorf tubes, and 1500 μL of the extraction solution composed of 41% water, 3% methanol, 12% acetone, and 43% acetic acid (*v*/*v*/*v*/*v*) was added. The tubes were stirred using an NA 3600 vortex mixer (Norte Científica, Araraquara, Brazil) to ensure the homogeneous suspension of the sample, and the extraction process was carried out exhaustively for six consecutive cycles (in duplicate). This extraction process involved low-frequency ultrasound for 5 min at room temperature (~20 °C), followed by centrifugation at 5000× *g* for 10 min at 20 °C, with the supernatants collected in a volumetric flask and the final volume adjusted to 10 mL with extraction solvent. Sanitization and soaking water have the volume adjusted to 250 mL. The Folin–Ciocalteu method was used to determine the absorbance of phenolic compounds. For the color reaction, 100 μL of the extract from raw material or breakfast cereals or 200 μL of sanitization and maceration water was added to 250 μL of Folin–Ciocalteu reagent (1:9 in distilled water), 3 mL of distilled water, and 1 mL of Na_2_CO_3_ 15%. The mixture was kept in a dark environment for 30 min to allow a complete color reaction. Absorbance readings were performed using an Anthos 200 ZT microplate reader spectrophotometer (Zenyth, São Paulo, Brazil) at 750 nm. The 7-point standard curve was constructed using gallic acid (0 to 600 mg·L^−1^; y = 0.0008x − 0.0019; r = 0.9979).

#### 2.2.6. Statistical Analysis

The normality of the data was checked using the Shapiro–Wilk test. The differences in centesimal composition between USR and SR, SR and MF, and USR and MF were determined using the Student’s *t*-test (*p* < 0.05). Analyses conducted for WAI and WSI, instrumental color, and TSPC of flours, as well as for the responses of breakfast cereal trials (REI, BD, instrumental texture, GABA, and TSPC), were evaluated by one-way analysis of variance (*p* < 0.05), assuming a Gaussian normal distribution and homogeneity of variances. When significant differences were observed, the Scott–Knott test was applied to detect differences among the samples.

## 3. Results and Discussion

### 3.1. Nutritional, Technological, and Physicochemical Properties of Raw Material

The proximate composition (Table 1) of MF, SR, and USR used to prepare breakfast cereals includes digestible carbohydrates, total dietary fibers, proteins, ash, and ether extract.

The studied whole flours had moisture values below 15%, complying with the requirements stipulated for the suitable moisture content of cereals and vegetable flours, ensuring the quality and safety of the flours [20].

The data obtained for the proximate composition confirm that SR had a higher protein and mineral content than MF but lower levels of ether extract and total carbohydrates (*p* < 0.05). In the case of USR compared to MF, significant differences were observed for all the components (*p* < 0.001). Significant differences in digestible carbohydrates were observed between SR and USR (*p* < 0.05).

The SR and USR flours showed no statistically significant difference in protein content (*p* = 0.630). It’s important to note that protein analysis involves quantifying total nitrogen, which includes all nitrogen compounds (condensed, bound, and free), especially the free amino acids produced during the germination process for SR. However, notable differences were observed for the other components (*p* < 0.05). Regarding ash content, SR exhibited lower values, which can be attributed to leaching during the sanitization, maceration, and washing of grains that underwent the germination process. In this case, the primary reduction occurred in free and monovalent minerals due to their higher solubility in water. There was also a decrease in dietary fiber content in SR compared to USR. This reduction may be linked to the partial breakdown of the fiber-rich cell wall, as the germination process promotes the activity of cell wall-degrading enzymes (esterases, phytases, tannins, cellulases, and hemicellulases), leading to hydrolysis that provides energy for radicle development. The partial hydrolysis of dietary fiber likely contributed to the increase in quantified digestible carbohydrates in SR, as the analytical method used to assess digestible carbohydrates involved combined gelatinization (heating in the presence of an alkali) and subsequent acid hydrolysis. As some dietary fibers can be hydrolyzed during sprouting, these low-molecular-weight fractions may have undergone hydrolysis and been measured. It is known that interactions occur between polysaccharides and the interaction of carbohydrates with ions, proteins, and lipids, which aligns with the results observed in this study regarding the ether extract. Without the germination process, complete extraction of all nonpolar components (lipids, pigments, vitamins, waxes, and phytosterols) may not have occurred. Still, after sprouting, the action of endogenous enzymes enhanced the extraction of hydrophobic components. Germination of cereal grains results in increased enzymatic activity, total dry matter loss, and substantial changes in nutritional composition.

Initially, the technological properties of the raw materials (Table 2) were evaluated concerning the water absorption index (WAI) and water solubility index (WSI). According to El Sohaimy et al. [21], WAI and WSI are important technological properties as viscosity affects the texture conditions of expanded extruded products. The change in WAI and WSI in flours obtained from sprouted grains originates from the action of enzymes involved in the biotechnological process, including *α*-amylases, *β*-amylases, and amyloglucosidase, which partially hydrolyze starch, promoting starch hydrolysis, which is used as a carbon source for embryonic development during sprouting [22].

The WAI of MF was lower than that of SR and USR, and statistically, all the whole flours differed from each other (*p* < 0.001). The same behavior was observed for WSI (*p* < 0.001). SR exhibited the highest WAI, which may be related to the higher dietary fiber content in the sample, and, due to the sprouting process, soluble fibers favored hydration and viscosity development [11]. Additionally, SR had a higher WSI than flours made from unsprouted grains (MF and USR).

The instrumental color (Table 2) of the whole flours used was assessed by the parameters of brightness (*L**), red/green coordinate (*a**), and yellow/blue coordinate (*b**), with the colors resulting from the presence of natural pigments in the raw materials. Brightness ranged from 67.60 ± 0.04 to 77.37 ± 0.10. It was observed that the flours differed from each other (*p* < 0.001), with MF (Figure 2) showing a lighter shade and reddish (+*a**) and yellowish (+*b**) tones when compared to SR and USR.

The MF contains carotenoids (lutein and zeaxanthin) as natural pigments responsible for the yellow color of many foods [23]. In contrast, the SR and USR exhibited darker shades, which can be attributed to the high dietary fiber content in whole flours [16]. This is satisfactory, as consumers often associate darker colors with artisanal, whole, and healthier products [24]. For SR and USR, the main pigments are chlorophyll and carotenoids. SR showed higher values for the reddish (+*a**) and yellowish (+*b**) colors compared to USR (*p* < 0.001). During the sprouting process, the release of pigments esterified to dietary fibers and proteins may occur, favoring the bathochromic effect associated with the chromophore. Chlorophyll comprises a pyrrolic structure with central porphyrin rings and hydroporphyrin stabilized by a magnesium atom [25]. However, chlorophyll can be divided into a part that has a methyl group in its chemical structure and another that has a carbonyl at the primary carbon (aldehyde), making its structure more unstable due to greater polarity and electronegativity, resulting in a shift towards lower *a** values and a greener tone of chlorophyll with an aldehyde group compared to the methyl group. In most cases, chlorophyll is accompanied by carotenoids, which act as self-protection against oxidative stress and free radical scavenging. Carotenoids consist of carotenes (non-oxygenated) and xanthophylls (oxygenated), with nonpolar characteristics and tones ranging from yellow to red [26].

The sprouting technique induces the hydrolysis of macronutrients (starch, protein, dietary fiber, and lipids) and, at the same time, promotes the bioconversion of new health-promoting metabolites, notably phenolic compounds and GABA. According to Table 2, GABA values ranged from 4.37 ± 0.02 to 40.83 ± 3.78 mg.100 g^−1^ (d.b.) for the flours. It was observed that USR and MF did not differ in terms of GABA content. On the other hand, the sprouting process increased the GABA content in ryegrass by 6.7 times (*p* < 0.001). These results confirm that sprouted seeds are rich sources of GABA compared to non-sprouted grains [3]. This effect can be attributed to the activation of seed metabolism, particularly the activation of glutamate decarboxylase (GAD) during the sprouting process, which favors the conversion of glutamate into GABA. The transferase enzyme can also be activated, resulting in increased glutamic acid, which is converted into GABA [11].

Phenolic compounds are secondary metabolites that commonly feature a hydroxylated aromatic ring. They act as a plant defense against biotic and abiotic factors or when subjected to stress conditions such as infections, pests, mechanical injuries, and radiation. In foods, phenolic compounds are responsible for color, astringency, aroma, and oxidative stability [27]. The TSPC content in SR and USR flours was statistically higher when compared to MF and differed from each other (*p* = 0.001). When we look at the values obtained for SR compared to USR, we can infer the action of endogenous enzymes such as carbohydrases, proteases, lipases, phytases, tannases, cellulases, and hemicellulases, which are synthesized during the sprouting process. These enzymes promote the release of phenolic compounds from their bound forms to their free forms and also contribute to the degradation and/or bioconversion of macromolecules present in the raw material [28].

It was observed that SR suffered a loss of TSPC through leaching into the sanitization and soaking water. Therefore, the quantification of TSPC for sanitization water was 29.75 mg GAE·100 g^−1^ (d.b.), and for soaking water, it was 20.56 mg GAE·100 g^−1^ (d.b.), representing a loss of approximately 5.98% of TSPC through leaching regarding the phenolic compounds in ryegrass seeds. Considering these losses, the sprouting process was conducted using ryegrass seeds with an initial content of 791.15 mg GAE·100 g^−1^, d.b. However, on the other hand, SR showed a 35.82% increase in TSPC achieved through the sprouting process compared to the content after sanitization and soaking. Furthermore, the isolation, identification, and characterization of bioactive compounds, followed by an appropriate extraction process, are only possible. The phenolic compounds’ extraction process involves the use of different solvents. Phenolic compounds are characterized as diverse substances that vary in chemical structure (aromatic rings and organic carboxylic acids), the number of functional chemical groups (hydroxyls), and distinct polarities. Therefore, solvents with different polarities should be employed, with water having a relative polarity of 1, while methanol, acetone, and acetic acid possess relative polarities of 0.769, 0.365, and 0.648, respectively. The appropriate mixture of the extraction solution enhances the extraction of TSPC present in ryegrass [29]. Therefore, solvent selection is one of the most crucial parameters in recovering the target compound.

Kagan et al. [30] reported that the type of raw material shows differences in profile and concentration of phenolic compounds between cultivars. A comparison of two perennial ryegrass (PRG) cultivars was investigated by the authors, where “Calibra” (PRG) showed higher values of 1210 mg GAE·100 g^−1^, compared to “Linn” (PRG), which had values around 970 mg GAE·100 g^−1^ using an extract solution composed of methanol 60%, water 39%, and acetic acid 1%.

### 3.2. Technological Properties of Breakfast Cereals

#### 3.2.1. Radial Expansion Index and Bulk Density of Breakfast Cereal

The REI assesses the physical characteristics of the extruded material related to the size, quantity, and distribution of air cells within it. Expansion occurs when the viscous material exits the extruder’s die [11]. After expansion, there is a slight collapse of the material until the structure stabilizes when it reaches the glass transition with a sharp temperature reduction. The higher the REI, the greater the material’s porosity, and the lower the BD, representing the weight the structured extrudate occupies in a given volume. The values ranged from 1.93 ± 0.15 to 2.50 ± 0.09 for REI and from 0.13 ± 0.02 to 0.18 ± 0.03 (g·cm^−3^) for BD (Table 3).

By the Scott–Knott test (*p* < 0.05), REI showed a similarity between SR4 and USR10, with higher values. The assays with higher percentages of SR (SR16 and SR20) substituting MF presented the lowest REI values. In contrast, the assay with the lowest addition of SR (SR4) showed a higher REI, differing statistically from the other trials with SR (*p* < 0.001) but with a similar behavior to URS10. This effect can probably be explained by the increase in protein and dietary fiber contents with SR since these macrocomponents hinder extrudate expansion by promoting a more dense material that is resistant to flow and capable of physically rupturing air bubbles during growth.

Additionally, the action of endogenous enzymes from the sprouting process acts on the starch, forming molecules with a lower molar weight. This also favors the reduction of the medium’s viscosity since a lower molar weight promotes an increase in solubility, as indicated by the WSI for SR in Table 2 [9]. SR0 differed statistically from all other assays (with SR4 to SR20 and USR10), presenting a lower REI value. The expansion of breakfast cereals depends on the size of starch granules and the length of the amylopectin molecule, knowing that amylopectin has a high level of ramifications (α-1→6 of glycosidic bounds) that contribute to high molar weight, possibly generating an increase in the degree of expansion [13]. Starch is the best material to undergo the thermoplastic extrusion process for expanded extrudates. In the case of SR0, elaborated only with MF, it was observed in Table 2 that it has a high lipid content (6.27%), which results in lubrication within the extruder barrel, reducing friction and shear, resulting in a lower degree of starch thermodextrination, an essential characteristic for the occurrence of expansion. The harmful effect of MF lipids was minimized in the presence of small quantities of SR because there was partial substitution of MF, and with the presence of starch from SR, as well as proteins and dietary fiber in small proportions, possibly increased mechanical work inside the extruder barrel, favoring the REI with the use of 4% SR (SR4).

For the assay with USR (USR10), a similar effect on REI was observed because it is a flour obtained from non-sprouted grains, inferring that the hydrolysis of part of the starch may have resulted in lower viscosity inside the extruder, promoting a lower degree of expansion for SR4 to SR20. It is worth clarifying that measuring specific mechanical energy during the thermoplastic extrusion process would be an essential factor to help better understand this phenomenon. Still, the extruder used did not allow for torque recording during processing.

Regarding the BD of the extrudates, SR4 and USR10 differed from each other (*p* < 0.001). This characteristic may be related to the contribution of the sprouting process to SR, favoring starch gelatinization until extrusion cooking [11] since it has a higher WAI, reducing the energy required to disrupt starch granules and favoring the formation of polymer-water hydrogen bonds, consequently leading to the development of a viscoamorphous gel during the extrusion process. This gel likely facilitated better gas bubble retention, providing SR4 with a higher REI and lower BD than USR10. With this, we can observe that USR showed a more significant influence of fibers, affecting REI and resulting in higher BD. It is known that REI and BD are inversely proportional parameters, so larger air bubbles in the expanded product promote an increase in specific volume. The conditioning water acts as a plasticizer for the amorphous region of starch granules, contributing to the thermomechanical gelatinization process and favoring rheological properties and air bubble formation [31].

#### 3.2.2. Water Absorption Index and Water Solubility Index of Breakfast Cereals

The WAI of breakfast cereals (Table 3) ranged from 7.31 ± 0.14 to 7.71 ± 0.02 g of gel·g^−1^ of sample (d.b.). There was a trend toward reduced WAI for extrudates with higher percentages of SR since SR12, SR16, and SR20 were similar (*p* > 0.05). According to Hashimoto et al. [32], as the protein and fiber contents increase, the WAI of extrudates decreases. Dietary fibers are classified into insoluble and soluble fibers based on their solubility in water. Soluble fibers comprise non-cellulosic polysaccharides (*β*-glucans, some hemicelluloses, pectins, gums, mucilages, and others). Pectins and gums are easily hydrated, forming highly viscous gels [32], and thus, the contribution of soluble fibers led to a decrease in WAI in extrudates made with SR.

Regarding the WSI (Table 3), it reflects the amount of starch degraded into smaller molecules (dextrins) during extrusion through thermomechanical changes, such as the dietary fibers [33]. SR12 and SR20 were similar and showed the highest WSI values, probably due to the greater release of dextrins from SR. Dextrins, in turn, are more soluble in water and consequently promote higher WSI. In addition, endogenous amylolytic enzymes that act during sprouting promote partial starch fragmentation. Scanning electron microscopy (SEM) is commonly used to assess the action of enzymes on starch after sprouting. As observed by Andressa et al. [34], when comparing the control corn starch sample to sprouted corn starch, the control corn starch had intact starch granules with a hexagonal shape and small pores on the surface, commonly found in corn starch. However, SEM images of sprouted corn starch showed the presence of larger pores. This result can be attributed to the catabolism of endogenous enzymes, which hydrolyze the protein matrix and the *α*-1,4-D-glucosidic bonds of starch.

Sprouting occurs through processes of catabolism and degradation of the reserves (starch, proteins, and lipids) stored in the seeds to support the development of the radicle. [7]. Starch is hydrolyzed by the action of endogenous amylolytic enzymes into low-molecular-weight molecules such as glucose, maltose, maltotriose, maltotetraose, and maltodextrins [9]. Protein hydrolysis is driven by endogenous proteases, which release peptides, amino acids, and nitrogenous compounds, while lipases cleave triacylglycerides into free fatty acids [10]. Esterases and cellulases partially hydrolyze dietary fibers. Consequently, substantial changes in seeds’ chemical and biochemical composition occur, enhancing nutrients’ bioavailability and bioaccessibility [35]. Additionally, it is known that the extrusion process modifies the structural and chemical characteristics of foods, primarily through starch gelatinization and protein denaturation, affecting the microstructure and texture of the extrudate [12].

#### 3.2.3. Instrumental Color of Breakfast Cereals

The instrumental color of extruded products is a commonly used evaluation tool by consumers directly related to product acceptance and quality, consequently influencing consumers’ choices. Therefore, it is an important attribute to be analyzed. Additionally, color significantly impacts extruded products associated with the numerous reactions that occur during the extrusion process [16].

As observed in Table 4 and Figure 3, the *L** parameter (luminosity) was statistically significant (*p* < 0.001). The *L** values of the extrudates ranged from 50.29 ± 0.48 to 60.88 ± 0.28. At the same time, the non-extruded flours (SR, USR, and MF) had luminosity *L** values of 67.60 ± 0.04, 68.75 ± 0.04, and 77.37 ± 0.10, respectively, indicating color changes caused by the extrusion process. The extruded products containing levels above 8% of SR and with 10% of USR were darker (Figure 3 and Table 4) and did not differ from each other. 

The SR0 showed higher *L** values. This effect can be attributed to the layers of the maize kernel, which consist of pericarp and aleurone cells. In the case of the ryegrass grain, it includes a husk, pericarp, and aleurone cells. These layers consist of polysaccharides (cellulose, arabinoxylans, lignin, glucomannans, and *β*-glucan) that give the whole flour a brown appearance, which is even more noticeable for SR due to the contribution of the husk. Additionally, darkening can be explained by non-enzymatic browning reactions, such as Maillard and caramelization, occurring during extrusion.

Moreover, extruded products with a higher expansion index can appear visibly lighter due to the greater distance between the components of the raw materials that contribute color to the foods [13]. Significant differences were observed in the *L** parameter for SR4 compared to the other SR extrudates. This finding is consistent with the REI, as a more expanded product results in larger air cells, leading to higher *L** values. However, the *L** values for the SR8 to SR20 and USR10 samples did not show statistically significant differences. Lower *L** values are indicative of more compact air bubbles. Furthermore, the reduced expansion of breakfast cereals during the extrusion process resulted in greater entrapment of dietary fibers within the viscoamorphous mass formed, obscuring the influence of dietary fibers on the color of the final product [16].

For the reddish tone (*a**), higher values were observed in breakfast cereals with SR0 and SR4. For the coordinate *b**, a reduction was noticed in breakfast cereals obtained with higher levels of SR, with SR20 showing the lowest *b** value, and this effect was attributed to the degradation of carotenoids during the extrusion process [36,37] as by the MF replacement by SR. In the case of ΔE (Table 4), similarities were observed between SR4, SR8, and SR16. Similarities were also observed between SR12 and SR20, and USR10 differed statistically from all assays (*p* < 0.001).

#### 3.2.4. Total Soluble Phenolic Compounds and γ-Aminobutyric Acid for Breakfast Cereals

The whole grain flours studied here and the extrusion process significantly affected the phytochemicals, with a decrease in TSPC levels for SR4 (8.84%), SR8 (16.25%), SR12 (22.54%), SR16 (27.96%), SR20 (32.66%) regarding SR, and for USR10 (22.07%) regarding USR, based on the mass balance (*p* < 0.001). This reduction can be explained by thermal destruction or changes in their molecular structure promoted by the extrusion process. The TSPC levels (Table 5) ranged from 372.19 ± 28.04 to 699.36 ± 39.92 mg GAE·100 g^−1^ (d.b.). Regarding the experiments, SR20 had the highest TSPC content of 699.36 mg GAE·100 g^−1^ (d.b.), produced with 20% SR replacing MF, confirming the expected result for the flour obtained through sprouting.

There is relatively little information on the total phenolic concentrations of winter-grown grasses. According to Kagan et al. [38], the phenolic compounds in perennial ryegrass (*Lolium perenne* L.) are esterified in the cell walls and contain vacuole-soluble phenolic compounds. Kagan et al. [31] evaluated perennial ryegrass from the “Calibra” and “Linn” cultivars of the *Lolium* genus and identified in the extracts of Linn PRG and Calibra, 5-caffeoylquinic acid (neochlorogenic acid), 3-caffeoylquinic acid (chlorogenic acid), and 4-caffeoylquinic acid (cryptochlorogenic acid). The putative peak of 3-caffeoylquinic acid (retention time of 5.2 min) was about 10 times more abundant than the peak of 4-caffeoylquinic acid and the previously identified chlorogenic acid in grasses. It is worth noting that some groups were not identified due to the extraction method. Additionally, compound concentrations are influenced by genotype and cultivar origin [39] and environmental changes [40].

Regarding GABA in the experiments (Table 5), it was observed that USR had a higher GABA content compared to MF, demonstrating that ryegrass is an excellent candidate for food development due to its considerable GABA content. Furthermore, the sprouting process increased the GABA content in SR by 6.7 times (*p* < 0.001), making it a promising technique for increasing health-beneficial compounds.

In this context, the breakfast cereals presented significant variations, influenced mainly by the raw materials. Therefore, comparing the experiment prepared without the addition of sprouted ryegrass flour (SR0) (only with MF) with the experiments designed with the incorporation of SR at higher percentages (16 and 20%) showed a significant increase in TPC and GABA levels. The same trend occurs when comparing SR16 and SR20 with the 10% USR (USR10) experiment.

#### 3.2.5. Instrumental Texture for Breakfast Cereals

Texture is an attribute of great importance in extruded products. The breakfast cereals were evaluated for hardness (N), rupture force (N.s), and crispness both for dry breakfast cereals and in the bowl, referred to as bowl-life (Figure 4).

The hardness values (N) were statistically significant for dry breakfast cereals and texture in the bowl-life. It is worth noting that the values were not statistically significant for rupture force (N·s), and for crispness, the obtained values were statistically significant.

According to the Scott–Knott test (*p* < 0.001), USR had higher mean hardness and rupture force values for dry breakfast cereals. The bowl-life analysis observed the same trend, with higher mean hardness and rupture force values. Consequently, USR had lower crispness values for dry breakfast cereals and bowl-life analysis. Therefore, the breakfast cereal obtained with USR exhibited increased hardness and rupture force and a decrease in crispness, as shown in Figure 4. According to Menis-Henrique et al. [32], dietary fibers, especially insoluble fibers, tend to increase the hardness of extruded products due to the high fiber content of whole-grain flours.

Therefore, low hardness in breakfast cereals can be considered an essential attribute for their acceptability, as it will result in highly crisp extruded products, another characteristic of this product. The extruded breakfast cereal with minimal percentages of SR, such as SR8 (8%), showed better performance, with lower hardness and rupture force and higher crispness values both for dry breakfast cereals and for the analysis after immersion in whole milk (bowl-life).

## 4. Conclusions

The ryegrass grain has a higher content of dietary fiber and phenolic compounds and a lower amount of lipids and digestible carbohydrates compared to maize, making ryegrass an alternative to improve the nutritional quality of flours for use in cereal technology, such as in the case of breakfast cereals. SR and USR had superior nutritional value to MF. Thus, the use of the biotechnological sprouting process in ryegrass seeds proved to be interesting because the action of endogenous enzymes during sprouting resulted in significant increases in TSPC (1.19-fold) and GABA (6.73-fold) content, suggesting a possible increase in the bioavailability and bioaccessibility of macronutrients. Furthermore, breakfast cereals developed with SR showed adequate nutritional and physicochemical properties. From a technological standpoint, the expansion index and solubility index were improved with the partial replacement of MF by SR. From a healthy perspective, SR had higher amounts of dietary fiber, proteins, and bioactive compounds (TSPC, GABA). The sprouting process of ryegrass enhances the nutritional properties of the grains. Still, the application of flour resulting from the milling negatively impacted the technological properties of breakfast cereals when used in concentrations exceeding 8% concerning maize flour. This allows us to infer that the best formulation achieved was SR8. Therefore, these findings demonstrate that ryegrass can be an excellent alternative for developing healthy foods and/or as a source of bioactive compounds for food applications.

## Figures and Tables

**Figure 1 foods-12-03902-f001:**
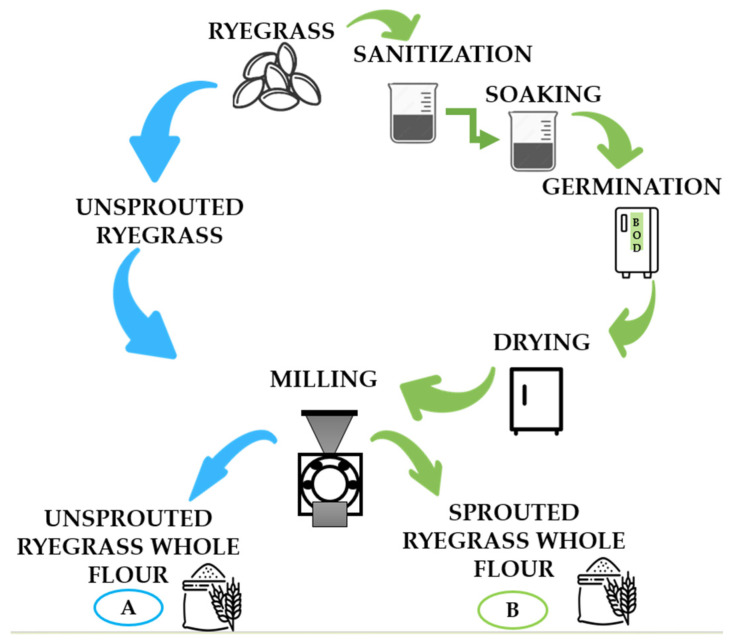
Flowchart of the process of obtaining the whole ryegrass flour. (**A**) Unsprouted whole ryegrass flour; ryegrass: whole ryegrass seeds; unsprouted ryegrass: unsprouted whole ryegrass seeds; and milling: grinding of unsprouted whole ryegrass seeds using a knife mill. (**B**) Sprouted whole ryegrass flour; ryegrass: whole ryegrass seeds; sanitization: disinfection of ryegrass seeds with a 250 ppm sodium hypochlorite solution for 30 min (1:6 *w*/*v*) (20 ± 2 °C); soaking: maceration of whole ryegrass seeds in distilled water (1:6 *w*/*v*) at 20 ± 2 °C for 4 h; germination: sprouting process of ryegrass seeds conducted in a BOD (20 ± 2 °C) for 95 h; drying: drying in an oven with forced air circulation and renewal (45 ± 2 °C) for 20 h; and milling: grinding of sprouted ryegrass seeds using a knife mill. The symbols were acquired from FlatIcon^®^, freely available at https://www.flaticon.com/br/ accessed on 30 August 2023.

**Figure 2 foods-12-03902-f002:**
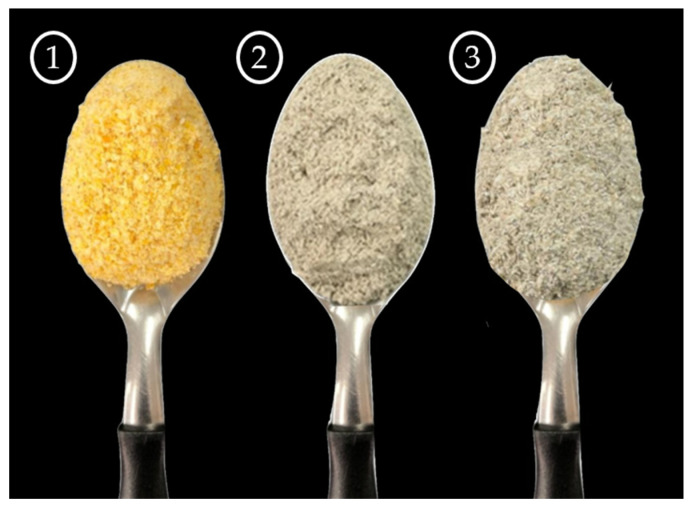
Visual appearance of the raw materials: 
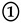
 whole maize flour; 
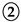
 whole sprouted ryegrass flour; and 
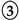
 whole unsprouted ryegrass flour.

**Figure 3 foods-12-03902-f003:**
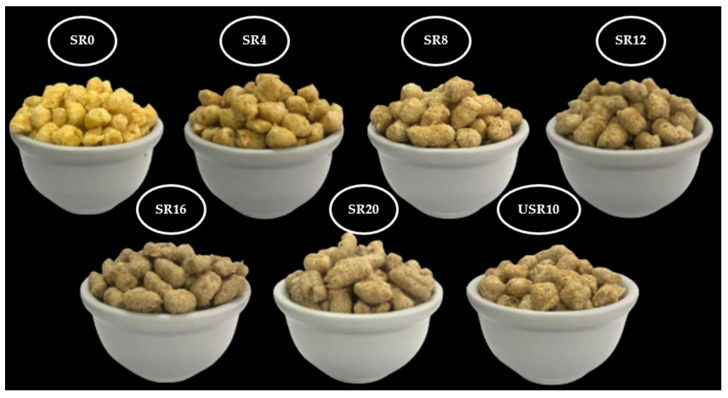
Visual representation of the breakfast cereals made with whole maize flour, sprouted whole ryegrass flour, and unsprouted whole ryegrass flour for each experimental trial is as follows: (1) SR0: without sprouted whole ryegrass flour (0%); (2) SR4: sprouted whole ryegrass flour (4%); (3) SR8: sprouted whole ryegrass flour (8%); (4) SR12: sprouted whole ryegrass flour (12%); (5) SR16: sprouted whole ryegrass flour (16%); (6) SR20: sprouted whole ryegrass flour (20%); (7) USR10: unsprouted whole ryegrass flour (10%).

**Figure 4 foods-12-03902-f004:**
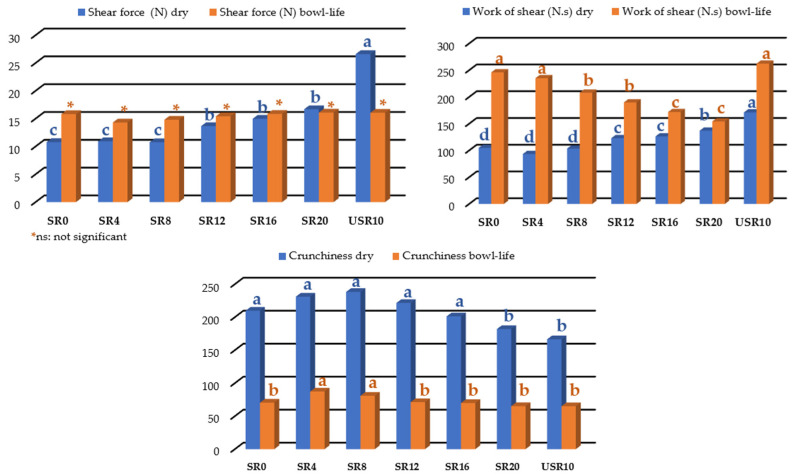
Instrumental texture data for the instrumental texture of dry and bowl-life breakfast cereals produced from whole maize flour, whole sprouted ryegrass flour, and whole unsprouted ryegrass flour. Different letters above bars of the same color indicate statistical significance (*p* ≤ 0.05). SR0: sprouted whole ryegrass flour (0%); SR4: sprouted whole ryegrass flour (4%); SR8: sprouted whole ryegrass flour (8%); SR12: sprouted whole ryegrass flour (12%); SR16: sprouted whole ryegrass flour (16%); SR20: sprouted whole ryegrass flour (20%); USR10: unsprouted whole ryegrass flour (10%).

**Table 1 foods-12-03902-t001:** Chemical composition of raw material (g·100 g^−1^, on a dry basis).

Component	Maize Flour	Sprouted Ryegrass	Unsprouted Ryegrass
Proteins	7.03 ± 0.50	13.12 ± 0.16	13.11 ± 0.37
Ether extract	6.27 ± 0.05	2.70 ± 0.10	2.15 ± 0.18
Ashes	1.51 ± 0.68	5.09 ± 0.02	5.78 ± 0.01
Digestible carbohydrates (starch and sugars)	80.19 ± 0.64	65.17 ± 0.56	63.69 ± 0.43
Total dietary fibers *	5.00 ± 0.87 ^#^	13.92 ± 0.77 ^#^	15.27 ± 0.35 ^#^

Data are the means ± standard deviation of three replicates (n = 3). * Total dietary fiber content calculated by the difference [100—(proteins + ether extract + ashes + digestible carbohydrates)]. ^#^ Standard deviation calculated by error propagation.

**Table 2 foods-12-03902-t002:** Physico-chemical parameters and nutritional composition of raw materials.

Parameters		Maize Flour	Sprouted Ryegrass	Unsprouted Ryegrass
Water absorption index (g of gel·g^−1^ of sample, d.b.)		2.75 ± 0.01 ^c^	3.82 ± 0.02 ^a^	3.48 ± 0.17 ^b^
Water solubility index (%, d.b.)		5.31 ± 0.07 ^c^	11.72 ± 0.19 ^a^	6.80 ± 0.21 ^b^
Instrumental color	*L**	77.37 ± 0.10 ^a^	67.60 ± 0.04 ^c^	68.75 ± 0.04 ^b^
	*a**	7.28 ± 0.01 ^a^	2.72 ± 0.01 ^b^	2.57 ± 0.01 ^c^
	*b**	39.74 ± 0.05 ^a^	14.75 ± 0.01 ^b^	14.43 ± 0.02 ^c^
γ-aminobutic acid (mg·100 g^−1^, d.b.)		4.37 ± 0.02 ^b^	40.83 ± 3.78 ^a^	6.07 ± 0.04 ^b^
TSPC (mg GAE·100 g^−1^, d.b.)		537.19 ± 31.48 ^c^	1002.58 ± 37.67 ^a^	841.46 ± 14.56 ^b^

Data are the means ± standard deviation of three replicates (n = 3). Different letters in rows denote statistical differences among raw materials (ANOVA, Scott–Knott, *p* ≤ 0.05). Abbreviations: d.b., dry basis; TSPC, total soluble phenolic compounds; GAE, gallic acid equivalents.

**Table 3 foods-12-03902-t003:** Technological properties and physicochemical characterization of breakfast cereals produced by replacing maize flour with whole sprouted ryegrass flour or whole unsprouted ryegrass flour.

Trials	Radial Expansion Index	Bulk Density (g·cm^−3^)	Water Absorption Index (g of gel/g of Sample, d.b.)	Water Solubility Index (%, d.b.)
SR0	1.93 ± 0.15 ^d^	0.18 ± 0.03 ^a^	7.71 ± 0.02 ^a^	25.68 ± 1.05 ^b^
SR4	2.36 ± 0.13 ^a^	0.13 ± 0.02 ^c^	7.31 ± 0.14 ^b^	25.28 ± 1.95 ^b^
SR8	2.30 ± 0.10 ^b^	0.13 ± 0.01 ^c^	7.59 ± 0.30 ^a^	24.48 ± 0.75 ^b^
SR12	2.05 ± 0.08 ^b^	0.15 ± 0.02 ^b^	7.42 ± 0.03 ^b^	30.47 ± 1.50 ^a^
SR16	2.03 ± 0.09 ^c^	0.15 ± 0.01 ^b^	7.36 ± 0.05 ^b^	22.96 ± 0.77 ^b^
SR20	2.18 ± 0.08 ^c^	0.14 ± 0.01 ^c^	7.26 ± 0.07 ^b^	27.63 ± 1.15 ^a^
USR10	2.50 ± 0.09 ^a^	0.15 ± 0.01 ^b^	7.60 ± 0.09 ^a^	22.18 ± 1.22 ^b^
*p*-value	<0.001	<0.001	0.038	<0.001

Data are the means ± standard deviation of three replicates (n = 3). Different letters in the row denote statistical differences among raw materials (ANOVA, Scott–Knott, *p* ≤ 0.05, which represents the level of significance adopted for the study, and the *p*-value in the table is the actual assessed significance). Sprouted ryegrass flour (SR) partially replaced MF in proportions of 0 (SR0), 4 (SR4), 8 (SR8), 12 (SR12), 16 (SR16), and 20% (SR20), and by 10% of unsprouted ryegrass flour (USR) (USR10).

**Table 4 foods-12-03902-t004:** Instrumental color of breakfast cereals produced from whole maize flour, whole sprouted ryegrass flour, and whole unsprouted ryegrass flour.

Trials	*L**	*a**	*b**	ΔE
SR0	60.88 ± 0.28 ^a^	7.11 ± 0.11 ^a^	39.73 ± 0.22 ^a^	-
SR4	54.26 ± 1.42 ^b^	7.06 ± 0.14 ^a^	34.96 ± 0.39 ^b^	8.12 ± 1.34 ^b^
SR8	51.79 ± 1.25 ^c^	6.60 ± 0.14 ^b^	30.66 ± 0.61 ^c^	12.80 ± 1.32 ^b^
SR12	50.44 ± 0.53 ^c^	5.80 ± 0.05 ^d^	27.40 ± 0.39 ^d^	16.15 ± 0.61 ^a^
SR16	50.67 ± 0.37 ^c^	5.86 ± 0.06 ^d^	25.77 ± 0.21 ^e^	17.29 ± 0.22 ^b^
SR20	50.29 ± 0.48 ^c^	5.46 ± 0.10 ^e^	24.13 ± 0.11 ^f^	18.88 ± 0.30 ^a^
USR10	50.69 ± 0.70 ^c^	6.19 ± 0.20 ^c^	27.26 ± 0.30 ^d^	16.09 ± 0.38 ^d^
*p*-value	<0.001	<0.001	<0.001	<0.001

Data are the means ± standard deviation of three replicates (n = 3). Different letters in the row denote statistical differences among raw materials (ANOVA, Scott–Knott, *p* ≤ 0.05). Sprouted ryegrass flour (SR) partially replaced MF in proportions of 0 (SR0), 4 (SR4), 8 (SR8), 12 (SR12), 16 (SR16), and 20% (SR20), and by 10% of unsprouted ryegrass flour (USR) (USR10). *L** is the luminosity, *a** is the red/green coordinate; *b** is the yellow/blue coordinate; and ΔE is the total color difference.

**Table 5 foods-12-03902-t005:** Total soluble phenolic compounds and γ-aminobutyric acid content of breakfast cereals produced from whole maize flour, whole sprouted ryegrass flour, and whole unsprouted ryegrass flour.

Trials	Total Soluble Phenolic Compounds (mg GAE·100 g^−1^)	γ-Aminobutyric Acid (mg·100 g)
SR0	372.19 ± 28.04 ^f^	3.50 ± 0.10 ^e^
SR4	485.96 ± 17.31 ^e^	7.96 ± 0.55 ^c^
SR8	535.52 ± 44.56 ^d^	8.48 ± 0.71 ^c^
SR12	586.37 ± 45.82 ^c^	8.64 ± 0.41 ^c^
SR16	644.52 ± 34.75 ^b^	11.70 ± 0.60 ^b^
SR20	699.36 ± 39.92 ^a^	14.52 ± 0.97 ^a^
USR10	548.38 ± 30.18 ^d^	6.55 ± 0.20 ^d^
*p*-value	<0.001	<0.001

Data are the means ± standard deviation of three replicates (n = 3). Different letters in the row denote statistical differences among raw materials (ANOVA, Scott–Knott, *p* ≤ 0.05). Sprouted ryegrass flour (SR) partially replaced MF in proportions of 0 (SR0), 4 (SR4), 8 (SR8), 12 (SR12), 16 (SR16), and 20% (SR20), and by 10% of unsprouted ryegrass flour (USR) (USR10).

## Data Availability

All the data presented in this study are available in this article.

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
