# Peer review of "New Breakfast Cereal Developed with Sprouted Whole Ryegrass Flour: Evaluation of Technological and Nutritional Parameters"

_foods, 2023, doi:10.3390/foods12213902_

Round 1

Reviewer 1 Report

IMPRESSION OF THE ARTICLE

The article investigates technological and nutritional characteristics of maize flour and (sprouted) whole ryegrass and breakfast cereals made thereof. Although the development of breakfast cereals with sprouted ryegrass is novel and the obtained results show that those cereals could have an added value for the food industry, the overall quality of the manuscript should highly be improved before it is acceptable for publication. Academic English, especially in the introduction, should be improved (i.e. word order, structure of sentences, choice of words,…), not all comparison to literature is relevant, figures in the results section do not have additional value, the discussion section should be better structured and statements should be made with reference to correct literature. In addition, the results & discussion section contains redundant information not directly related to the research topic and should be made more concise. I therefore recommend to reject the manuscript in its current state and after highly improving its quality allowing resubmission to the current or another Journal. I also kindly refer to comments below.

OVERALL COMMENTS

1.      There are a lot of mistakes against the English and scientific language. Please let someone with native tongue correct the manuscript.

2.      Numbering of the materials & methods and results & discussion section are not correct.

3.      SR0 =  MF, while both terms are used which is confusing to the reader. Please use only one term.

4.      Be precise which samples you are comparing, for example ‘a higher WSI compared to which other sample?’

5.      The effect of activated and de novo synthesized enzymes (e.g. amylases) during sprouting is often the major issue for detrimental quality when extruding whole meal from sprouted grains and should be discussed throughout the manuscript.

6.      10% of the MF was replaced with USR, but no sample was made where 10% of MF was substituted with SR. Therefore, it is not possible to compare USR with SR.

7.      As breakfast cereals are mostly flaked, toasted and/or deep-fried after extrusion, is it relevant to analyse only an intermediate product and not a final product?

SPECIFIC COMMENTS

TITEL

Performance of what? On technological and nutritional parameters of what?

ABSTRACT

Improve the clarity and structure of the sentences. There is nothing mentioned on λ-aminobutyric acid and TSPC of the breakfast cereals in the abstract, whereas this is most important in the purpose of producing more healthy breakfast cereals.

19-21                   Improve the clarity of this sentence as the message is unclear and avoid repetition of the word ‘viable’.

21                         The presence of bioactive compounds has also a nutritional impact, so why mention both?

25                         The bio-availability of aminobutyric acid has not been examined.

27-28                   The effect of activated and de novo synthesized enzymes (e.g. amylases) during sprouting is often the major issue for detrimental quality when extruding whole meal from sprouted grains.

28-29                   breakfast cereals with 4% and 8% SR exhibited better physicochemical properties compared to?

30-31                   There was only a significant reduced hardness (in bowl life cereals) in SR8 and not in SR4 compared to SR0. The increased crispness was in bowl-life cereals and not in dry cereals à Please be more precise.

31-32                   Bioactive compounds influence the nutritional quality, so why mention both? What about the technological parameters?

INTRODUCTION

Academic English should be improved (i.e. word order, structure of sentences, choice of words,…).  Authors should re-evaluate which information is crucial for understanding the obtained results & discussion and structure this in a comprehendible way.

36                         Ready-to-eat is an accepted term. Please look at the structure of sentences.

36                         gaining market share
37-38                   allegation: not the correct choice of word (allegation is a claim or assertion that someone has done something illegal or wrong, typically one made without proof)

38                         What is meant with ‘lack of time by consumers’ à Please be more clear.

43-45                   Please look at the structure of sentences.

45-46                   Please explain in more depth.

46-49                   Not all bioactive compounds will sequester free radicals. “Pathological condition” is not a correct choice of word as this has a negative allegation whereas the message is that bioactive compounds are health promoting.

50                         The use of “we have” can be made more scientifically precise. Please use passive voice or rephrase.

53                         The research topic is grains and not vegetables.

55                         ‘In this context’ is not a correct choice of words à please rewrite

55                         ‘belonging’ instead of ‘belongs to’

61                         Please add a reference.

62                         The use of “we have” can be made more scientifically precise. Please use passive voice or rephrase.

62                         What is meant with a millennial process? Please clarify.

69                         The terms sprouting and germination are used interchangeable which can be confusing for the reader.

88                         The formation of the amorphous paste is not the cause of starch gelatinization and protein denaturation

90-92                   Those 2 sentences are not relevant to this study as the study referred to starts from 100% maize flour.

MATERIALS AND METHODS

The numbering of the sections is not correct. There are two 2.2.3. numberings. It makes more sense to change the order of the sections (first current 2.2.2, then 2.2.3 and then 2.2.1) as the germination (2.2.2) occurs first, then the milling (2.2.3) and then the analysis of the composition (2.2.1). The structure and order of the paragraphs should be improved as currently the messages brought are unclear for the reader. Additionally, not all materials used in the study are listed in this section.

103                       Not all materials used to perform the experiments (solvents etc.) are listed here.

107-109              This sentence is not needed in materials & methods.

118-119              Are the results expressed as g/100g of dry matter?

121                       What was the volume of NaClO used for the sanitization of the seeds?

126                       How many seeds were placed in one tray?

137-139              The maize flour is sieved with a 0.5 mm sieve, whereas the ryegrass flour is sieved with a 0.3 mm sieve;  à why?

140-151              The figure has no added value to the article as the details are already given in 2.2.3.

149                       No captical letter is needed in ‘Grinding of germinated ryegrass’.

157                       The replacement 0%-20% à is this on volume or mass base?

159                       10 % of the MF was replaced with USR, but no sample was made were 10% of MF was replaced with SR. Therefore, it is not possible to compare USR with SR.

166                       Was water added to the whole flour during extruding and if so, how much?

167                       Was the pressure build-up in the extruder measured and the same for all the samples?

171                       As breakfast cereals are mostly flaked, toasted and/or deep-fried after extrusion, is it relevant to analyse only an intermediate product and not a final product?

171                       Were the extrudates milled before the analyses of REI, BD etc. ?

175                       Expansion index is related to the bulk density à Has it an added value to give both parameters?

175                       instrumental texture of dry and bowl-life à rewrite to ‘dry and bowl-life texture’

179                       the REI and BD were performed à rewrite to ‘analysis of REI and BD were executed according to’

180                       the ratio of extrudate à what parameters of the extrudate?

227                       10 min at 3,200 x g à remove the “x”

230                       Please be more precise on which analysis were done after the dehydration of the supernatans.

231                       the results are expressed in g of gel per g of sample à which gel?

232                       the results of WHI are expressed in percentage à percentage of what? Are the WAI and WSI related?

238                       ‘and centrifugated’ instead of ‘and centrifugation’

RESULTS AND DISCUSSION

The numbering of the sections is not correct: 3.2.6 is used before 3.2.5. The authors should focus more on the essence of the study: explain the most important results and refer to relevant literature. Currently secondary results are explained in too much detail leading to a lot of redundant information. There is regularly incorrect information given. References should be used to compare results and confirm hypotheses.

294                       results and discussion

295                       the parentheses in this sentence are not needed

297                       “as well as protein, ash and lipid contents”

297                       The terms digestible and non-digestible carbohydrates are used whereas in the table the term total dietary fibers is used for the non-digestible part à be consistent with the terms used

301                       Total dietary fiber content was calculated by difference à difference of what?

301                       Total dietary fibers à Why are the standard deviations calculated with error propagation?

304                       Please clarify this sentence as the message is not clear. ‘adequate’ is not a correct word of choice

312                       Is there a difference in condense, bound or free nitrogen compounds between the SR and USR?

315                       increase in dietary fiber content

316                       Is there cell wall biosynthesis during sprouting? Please clarify this sentence. What about dry matter loss (which can be seen in the increasing amount of digestible carbohydrates in sprouted ryegrass compared to unsprouted ryegrass)?           

317-319              This sentence should be in the introduction, not in the results & discussion.

321                       Remove ‘preserving’ à minimizing the detrimental effects but not preserving

326                       ‘Instrumental color’ should be a line higher in the table.

332                       The amount of soluble fiber has not been measured, so a reference should be added.

336                       The definition of starch dextrinization is “when treated only with dry heat or roasting, can undergo a molecular degradation process called dextrinization. This transformation can also take place in the presence of an acidic or alkaline catalyst.” So starch dextrinization does not occur during sprouting. What is the link between starch dextrinization and water solubility index? Please clarify.

336                       What is the content of starch in ryegrass? Will the starch dextrinization make a significant contribution to the water solubility index? Paragraph 338-343 seems to have more impact on the WSI, better to put this first and then the part on the starch dextrinization

343                       Additionally, starch undergoes partial hydrolysis à already mentioned in 336, so ‘additionally’ is not correct

344                       Which macro-components?

348                       This information should come earlier in the text.

356                       This picture has no added value to the article as the instrumental color values are given in table 2

360                       “SR and USR exhibited darker shades, which can be attributed to higher dietary fibre contents in the whole fours” à Can you confirm this sentence with a reference? Is it the dietary fiber that leads to the color or the bran of the grain?

369                       The structure is more unstable, leading to a greener ton à a greener ton compared to?

363                       What is the effect of sprouting on the instrumental color? 364-372 explains the effect of chlorophyll and carotenoids on the color, but the effect of germination on the color is not explained.

364-371              The information has no added value to the results, so this paragraph can be removed.

377                       Is this result also confirmed in other studies?

391                       ‘SR concerning the other flours’ à only compared to USR, not compared to MF

397                       ‘In summary,’ à the information given in this sentence is new, so starting the sentence with in summary is not correct.

400                       5.98% à is this percentage the total (sanitization and soaking water) loss due to leaching. Is this percentage against the total TSPC from the flour?

401                       What is meant with ‘a sample’ in this sentence: The germination process was conducted using a sample with 791.15 mg GAE*100g-1? Please clarify this sentence.

402                       35.82% à against the total TSPC in the raw material? Or against the 791.15 mg GAE*100g-1?

404                       What is meant with the sentence: “Furthermore, the isolation, identification, and characterization of bioactive compounds followed by an appropriate extraction process are only possible.” Please clarify.

405-407              Is this relevant?

408                       Crucial factor for what?

409                       The cultivar promotes changes à when are those changes happening?

412                       Why did you use another extract solution?

416-417              Please refer to literature in this sentence.

423                       Is it necessary to show both REI and BD as these parameters are linked to each other?

425                       higher values compared to?

426                       lowest REI values compared to? SR0 is lower than SR16 and SR20.

427                       differing statistically from others à which are the others, USR10 has same expansion index as SR4

449                       The use of MF and SR0 (which is only MF) is confusing. Please use only one abbreviation.

449                       10% of the MF was replaced with USR, but no sample was made where 10% of MF was substituted with SR. Therefore, it is not possible to compare USR with SR.

432                       Please be more precise on how a decrease in viscosity result in differences in WSI.

434                       Other assays à What is meant with this? Other breakfast cereals?

434-435              ‘knowing that amylopectin has ramifications that contribute to high molar weight’ à The message is unclear, please clarify.

443                       By replacing only 4% of MF with SR, the amount of MF lipids is not reduced as much. Additionally, the amount of starch in SR is limited (less than in MF) and you state that starch is important for expansion.

444                       In 428 you state that proteins and dietary fibers have detrimental effect on expansion, and in sentence 444 you state that proteins and dietary fiber would increase the REI.

447                       repetition of sentence 429 à not needed

451                       In the caption a p<0.05 is given, whereas in the table other p-values are given. What is the difference?

454                       similar effect as which effect?

455                       hydrolysis of part of starch à in non-sprouted grains? You can not compare USR10 with another SR, as no SR10 was used. Please clarify this sentence as the message is not clear.

458                       There is a lot of literature on extrusion of unsprouted cereals, so specific mechanical energy measurements during extrusion may not be necessary.

461-462              This sentence should be closer to 428 as in this sentence the effect of protein and dietary fiber is mentioned for the first time.

463                       As Hashimoto et al. uses cowpea cotyledon and no maize flour or ryegrass, the expansion index will be different due to the difference in composition. The higher expansion index value is not only due to the hull and the reduced dietary fiber content.

468                       How much of the starch is gelatinized during sprouting? Please add references.

469                       What is the link between starch gelatinization and gas bubble growth?

470                       In the text is stated that SR4 has a higher REI than USR10, whereas in the table this is the opposite.

470                       ‘with USR’ can be removed

472                       REI and BD are inversely proportional parameters, so why do we see a difference in BD between SR4 and USR10 and no difference between both in REI? Is it useful to show both BD and REI? Does giving both parameters have an added value to the article?

473-476              Is this information relevant to the results?

480                       Hashimoto used cowpea cotyledon flour and no cereals. Therefore it is possibly that the conclusion of his research is not applicable to other products like cereals.

487                       bioavailable is not the correct term as this is not evaluated. Lower values of what?

489                       The amount of starch degraded à only during extrusion? Or can this also happen during soaking and sprouting? Are there also other components influencing WSI?

494-499              This paragraph has no added value to the result section as no explanation on the differences in WSI in extrudates is given.

490                       Why isn’t there a higher WSI in SR16?

502                       Results on the instrumental color of the raw material are already given in a previous part of the article, so this sentence on instrumental color should come in that part and not here.

506                       Color can affect color? Please clarify.

507                       Please refer to table 4 and figure 3.

514                       Please be more clear why SR0 has a higher L* value. Why do we see a difference between SR4 and SR8-20 and no difference between SR8 to SR20?

517                       The husk is not edible. Is the husk used during breakfast cereal production?

521                       Please refer to literature to confirm the hypotheses.

522                       Has this figure an added value to the article?

530                       In the table Delta E is given, whereas this parameter is not given in table 2 or explained in materials & methods. Please do so if this parameter has an added value to the article. If not, please remove the parameter from the table.

536                       This sentence is not clear. Please rewrite.

544-560              This paragraph is not needed here. This information should be in the introduction section.

562                       Why is the data for SR0 lacking?

599                       Comparison with literature?

615                       *na: nao significativo à Pleas rewrite in English

see above

Author Response

We acknowledge the reviewers’ suggestions regarding our manuscript titled “Newly breakfast cereal developed with sprouted whole ryegrass flour and performance on technological and nutritional parameters”. Therefore, we have detailed the responses to the questions raised by the reviewers below (in red color). In addition, all the changes we have introduced into the revised manuscript are highlighted through “Track Changes”. This makes us confident that you may now find our manuscript suitable for publication in Foods.

Reviewer 2 Report

The effect of germination on the characteristics of ryegrass and the performance of germinated ryegrass whole flour on the physicochemical, technological, and nutritional properties of ready-to-eat breakfast cereal was studied in the manuscript entitled “Newly breakfast cereal developed with sprouted whole ryegrass flour and performance on technological and nutritional parameters” (Manuscript ID: foods-2633594) submitted to Foods. This study is interesting to meet the demand for nutritional food made with non-wheat flour, especially in regions where wheat is hard to cultivate. While the manuscript was well designed, a few issues should be addressed. The main concerns:

  1. There is too much information contained in the introduction. It is better to reorganize it to make the background and novelty of this study more concise and clear.
  2. Some logical connectives are not used properly, which makes the sentences not conducive to understanding. For example, the “However, ” in Line 82 is not appropriately used.
  3. Project information in Line 107-109 should not in the Materials. It may be in another part of the manuscript, for example, the Fundings or the Acknowledgement.
  4. L135 The title is incomplete. It should be “Obtaining the whole flours of unsprouted ryegrass, sprouted ryegrass and maize. Moreover, it is better for the authors to explain why different sieves were used briefly.
  5. In Table 3, it is weird that USR10 was used for comparison when SR 10 was not studied here. 
  6. The footnote in Figure 4 needs to be in English.

The English language is fine in this Manuscript. 

Author Response

(The authors gave the same response as above.)

Reviewer 3 Report

Please look at the attachment

Author Response

(The authors gave the same response as above.)

Round 2

Reviewer 1 Report

foods-2633594: Newly breakfast cereal developed with sprouted whole ryegrass flour and performance on technological and nutritional parameters

Foods

The author has revised the manuscript thoroughly. The academic and scientific language in the  introduction section has been improved. However, mistakes against the English language are still present. The discussion and results section contains redundant information not directly related to the research topic and should be made more concise. Please find the comments below, marked in yellow. I therefore recommend to reconsider this manuscript after major revisions.

SPECIFIC COMMENTS

ABSTRACT

21                         The presence of bioactive compounds has also a nutritional impact, so why mention both?

The request was accepted, and the sentence was rewritten in the abstract.

The study does not evaluate the nutritional impact of γ-aminobutyric acid and bioactive compounds. It does evaluate the content of both in the newly developed cereals. Please rewrite this sentence.

31-32                   Bioactive compounds influence the nutritional quality, so why mention both? What about the technological parameters?

The request was accepted, and the sentence was rewritten in the abstract.

Please be more precise on which technological parameters were maintained.

INTRODUCTION

88                         The formation of the amorphous paste is not the cause of starch gelatinization and protein denaturation

The request was accepted, and the sentence was rewritten.

This sentence is still uncorrect. The ‘such as’ is not correct in this sentence. Starch gelatinization and protein denaturation are the processes that lead to the viscous and amorphoud paste

321                       Remove ‘preserving’ à minimizing the detrimental effects but not preserving

MATERIALS AND METHODS

137-139              The maize flour is sieved with a 0.5 mm sieve, whereas the ryegrass flour is sieved with a 0.3 mm sieve;  à why?

The request was accepted, and the information was included in the text.

“The difference in sieve openings occurred because they raw materials were ground in different mills and countries, allowing the use of the available sieves.”

Please change ‘they’ into ‘the’.

171                       As breakfast cereals are mostly flaked, toasted and/or deep-fried after extrusion, is it relevant to analyse only an intermediate product and not a final product?

We would like to thank the reviewer for allowing us to clarify the production process of the breakfast cereal referred to in this study. Breakfast cereals obtained through 3rd-generation extrusion require a unit operation for the final product to be ready for consumption. In our case, we utilized a 2nd-generation extrusion process, which enables the final product to be ready-to-eat without the need for further heat treatment, such as toasting, flaking, or frying.

Thank you for the clarification. Please mention this in the manuscript.

180                       the ratio of extrudate à what parameters of the extrudate?

The sentence was rewritten for better understanding.

Please be more precise: ‘using the ratio of the diameter of the extrudate and die diameter’

RESULTS AND DISCUSSION

The authors should focus more on the essence of the study: explain the most important results and refer to relevant literature. Currently secondary results are explained in too much detail leading to a lot of redundant information.

312                       Is there a difference in condense, bound or free nitrogen compounds between the SR and USR?

Yes, because germination can promote the release of free amino acids.

Thank you for the clarification. Please mention this in the paragraph.

316                            Is there cell wall biosynthesis during sprouting? Please clarify this sentence. What about dry matter loss (which can be seen in the increasing amount of digestible carbohydrates in sprouted ryegrass compared to unsprouted ryegrass)?         

320                            A sentence on the importance of WAI and WSI is added, which is good. However, this sentence should come earlier in text (when the discussion on WAI and WSI starts)

336                       The definition of starch dextrinization is “when treated only with dry heat or roasting, can undergo a molecular degradation process called dextrinization. This transformation can also take place in the presence of an acidic or alkaline catalyst.” So starch dextrinization does not occur during sprouting. What is the link between starch dextrinization and water solubility index? Please clarify.

The term was adjusted to starch hydrolysis.

Please add reference 23 as it has been deleted. “Furthermore, enzymes involved in the sprouting process, including a-amylase …” The hydrolysis of starch is already mentioned in the previous paragraph. So the ‘furthermore’ is incorrect.

404                       What is meant with the sentence: “Furthermore, the isolation, identification, and characterization of bioactive compounds followed by an appropriate extraction process are only possible.” Please clarify.

This sentence refers to the appropriate use of solvents, either individually or in combination, to achieve the maximum extraction of TSPC (Total Soluble Phenolic Compounds).

Thank you for the clarification. Please rewrite this sentence as this sentence is not academically right and the message is not clear for the reader. Please be more precise on the result of the use of wrong solvents and why this is relevant to the results.

409                       The cultivar promotes changes à when are those changes happening?

In this information, we are referring to the difference between the cultivars.

Thank you for the clarification. The verb ‘promote’ may not be the correct choice as it indicates a change whereas it is the difference between cultivars and not a change in profile and concentration of phenolic compounds over time. 

434-435              ‘knowing that amylopectin has ramifications that contribute to high molar weight’ à The message is unclear, please clarify.

This sentence has been rewritten for better clarity.

Thank you. ‘possibly impact on and increase the degree of expansion.’. The ‘impacts on’ can be removed.

444                       In 428 you state that proteins and dietary fibers have detrimental effect on expansion, and in sentence 444 you state that proteins and dietary fiber would increase the REI.

This occurs because in small quantities, proteins and fibers lead to an increase in the torque required for the extrusion process.

Thank you for the clarification. Please remove the brackets in this sentence because the information that this is only in small amounts is crucial.

451                       In the caption a p<0.05 is given, whereas in the table other p-values are given. What is the difference?

P < 0.05 represents the significance level adopted for the study, and the p-value from the table is the assessed actual significance.

Thank you for the information. Please add this information to the caption of the table.

455                       hydrolysis of part of starch à in non-sprouted grains? You can not compare USR10 with another SR, as no SR10 was used. Please clarify this sentence as the message is not clear.

Thank you to the reviewer for pointing out this issue. However, we would like to mention that the germination of the grains was conducted in Brazil, and the extrusion cooking process was carried out in Peru. The limitation on the number of samples for our study was due to the quantity of ryegrass grains and the availability of materials to be transported between the countries. Therefore, for the purpose of comparison, we adopted the use of 10% USR, as it represented an intermediate level of substitution compared to the experiments with germinated flour (SR from 0 to 20%).

Thank you for the clarification. This sentence is confusing. Please rewrite to make the message more clear. By not sprouting the USR, the expansion of the sprouted ryegrasses is not ‘promoted’.

463                       As Hashimoto et al. uses cowpea cotyledon and no maize flour or ryegrass, the expansion index will be different due to the difference in composition. The higher expansion index value is not only due to the hull and the reduced dietary fiber content.

Thank you to the reviewer for pointing out this information. However, both raw materials are rich in dietary fibers and may have similar effects on the final product, thus aiding in the discussion of the results.

Thank you for the clarification. This paragraph has been moved to the section where the results of REI are discussed. However, the comparison between the results of Hashimoto and this study are not clear. Please add this information to explain the relevance of the comparison.

468                       How much of the starch is gelatinized during sprouting? Please add references.

This information was added to the sentence.

Is there literature on the percentage of starch that is gelatinized during sprouting? If so, please add a reference.

469                       What is the link between starch gelatinization and gas bubble growth?

This sentence has been rewritten for better clarity.

Not clear to what ‘it has a higher WAI’ refers to. Please be more precise. There is thus less energy required for the formation of polymer-water hydrogen bonds, but what is the link between this and the gel development? Please clarify.

473-476              Is this information relevant to the results?

Yes.

It is not clear to the reader why this information is relevant. Please be more precise.

489                       The amount of starch degraded à only during extrusion? Or can this also happen during soaking and sprouting? Are there also other components influencing WSI?

This sentence has been rewritten for better clarity.

Thank you for the clarification. A lot of information on the effect of sprouting on the formation of compounds is given. However, the information is not structured and redundant information is given. It is not clear why information on protein hydrolysis during sprouting is given just after the discussion of WSI. Please be more to the point.

502                       Results on the instrumental color of the raw material are already given in a previous part of the article, so this sentence on instrumental color should come in that part and not here.

The sections were separated to initially characterize the flours used in this study and, at this point, to evaluate the extruded product.

Thank you for the clarification. Please add ‘instrumental color of extruded products’ to avoid confusion.

514                       Please be more clear why SR0 has a higher L* value. Why do we see a difference between SR4 and SR8-20 and no difference between SR8 to SR20?

Because the similar values obtained S8-S20.

As more sprouted ryegrass is added, and thus more out layer of the ryegrass, the L* would increase between SR8 and SR20?

517                       The husk is not edible. Is the husk used during breakfast cereal production?

Indeed, cereal husks are not consumed. However, further studies should be conducted to determine if this can be changed.

Please remove the sentence ‘due to the contribution of the husk’ As this part is not included. Or was did part used during the extrusion and was the husk still present in the extrudants?

544-560              This paragraph is not needed here. This information should be in the introduction section.

The information presented justifies the importance of our study since there is insufficient data in the literature regarding ryegrass.

Not all information is needed to show the importance of the study and this information should be in the introduction section.

562                       Why is the data for SR0 lacking?

Sample SR0 was considered as the standard for the color analysis.

The percentages are they calculated based on the TSPC of SR0? It states that the the percentages are based on their respective unprocessed samples. Those samples are the flours? This is not clear for the reader.

Extensive editing of English language required

Author Response

We thank the reviewer for allowing us to further improve our manuscript titled “Newly breakfast cereal developed with sprouted whole ryegrass flour and performance on technological and nutritional parameters”. The title was changed to “Newly breakfast cereal developed with sprouted whole ryegrass flour: evaluation of technological and nutritional parameters”. Therefore, we have detailed the responses to the questions raised by the reviewers below (in green color). In addition, all the changes we have introduced into the revised manuscript are highlighted through “Track Changes”. Therefore, we hope that the article meets the quality standards required by Foods for the publication of our manuscript, which will be the first scientific publication on the potential use of ryegrass for human consumption.
